Combination of machine learning and data envelopment analysis to measure the efficiency of the Tax Service Office

Soffan Shofinurdin 1
Bramantoro Arif 2 arif.bramantoro@utb.edu.bn
http://orcid.org/0000-0002-8081-6530 Alzahrani Ahmad A. 3
1 Faculty of Information Technology, Universitas Budi Luhur , Jakarta , Indonesia
2 School of Computing and Informatics, Universiti Teknologi Brunei , Bandar Seri Begawan , Brunei
3 Faculty of Computing and Information Technology, King Abdulaziz University , Jeddah , Saudi Arabia
Cirillo Stefano
Electronic publication date: 2025 Feb 17
Publication date: 2025
Volume: 11
Electronic Location ID: e2672
Received 2024 Aug 1; Accepted 2025 Jan 7
Copyright: © 2025 Soffan et al.
Copyright year: 2025
Copyright holder: Soffan et al.
License: This is an open access article distributed under the terms of the Creative Commons Attribution License, which permits unrestricted use, distribution, reproduction and adaptation in any medium and for any purpose provided that it is properly attributed. For attribution, the original author(s), title, publication source (PeerJ Computer Science) and either DOI or URL of the article must be cited.
License URL: https://creativecommons.org/licenses/by/4.0/

Keywords: Machine learning, Data envelopment analysis, Efficiency, Tax service office, Genetic algorithm, Multilayer perceptron, Fuzzy c-means.

Funding: Institutional Fund IFPIP: 1173-611-1443 Ministry of Education and King Abdulaziz University, DSR, Jeddah, Saudi Arabia This research work was funded by Institutional Fund Projects under grant no. (IFPIP: 1173-611-1443). This also received financial support from the Ministry of Education and King Abdulaziz University, DSR, Jeddah, Saudi Arabia. The funders had no role in study design, data collection and analysis, decision to publish, or preparation of the manuscript.

==============================
The Tax Service Office, a division of the Directorate General of Taxes, is responsible for providing taxation services to the public and collecting taxes. Achieving tax targets efficiently while utilizing available resources is crucial. To assess the performance efficiency of decision-making units (DMUs), data envelopment analysis (DEA) is commonly employed. However, ensuring homogeneity among the DMUs is often necessary and requires the application of machine learning clustering techniques. In this study, we propose a three-stage approach: Clustering, DEA, and Regression, to measure the efficiency of all tax service office units. Real datasets from Indonesian tax service offices were used while maintaining strict confidentiality. Unlike previous studies that considered both input and output variables, we focus solely on clustering input variables, as it leads to more objective efficiency values when combining the results from each cluster. The results revealed three clusters with a silhouette score of 0.304 and Davies Bouldin Index of 1.119, demonstrating the effectiveness of fuzzy c-means clustering. Out of 352 DMUs, 225 or approximately 64% were identified as efficient using DEA calculations. We propose a regression algorithm to measure the efficiency of DMUs in new office planning, by determining the values of input and output variables. The optimization of multilayer perceptrons using genetic algorithms reduced the mean squared error by about 75.75%, from 0.0144 to 0.0035. Based on our findings, the overall performance of tax service offices in Indonesia has reached an efficiency level of 64%. These results show a significant improvement over the previous study, in which only about 18% of offices were considered efficient. The main contribution of this research is the development of a comprehensive framework for evaluating and predicting tax office efficiency, providing valuable insights for improving performance.

Introduction

The taxation sector in Indonesia plays a crucial role, being the primary contributor to state revenue. In 2022, revenue from taxation amounted to IDR 2,034.54 trillion, accounting for 77.5% of total state revenue (Ministry of Finance of the Republic of Indonesia, 2023). However, the tax ratio remains relatively low at 10.1% of the gross domestic product, which is lower than the average tax ratio of Asia Pacific countries (19%) and the OECD tax ratio (33.5%) (OECD, 2022). To improve the tax ratio, the Indonesian government has consistently pursued policies aimed at increasing tax revenue, as evidenced by the annual increment in tax revenue targets. For instance, the tax target rose from IDR 1,199 trillion in 2020 to IDR 1,718 trillion in 2023 (Directorate General of Taxes of Indonesia (DGT), 2021). Despite this drive for higher tax revenue, the growth of human resources in the taxation sector has not kept pace. In recent years, the number of tax employees has declined, with figures dropping from 46,607 in 2019 to 45,315 in 2022 (Directorate General of Taxes of Indonesia (DGT), 2020). This results in a low ratio of employees to taxpayers (1:7,742), far below the average ratio seen in OECD member countries (1:1,657) (Directorate General of Taxes of Indonesia (DGT), 2020). Addressing this situation requires the Directorate General of Taxes, responsible for tax collection, to function effectively and efficiently with existing resources. Their vision of becoming a trusted partner in national development through efficient, effective, integrity-based, and fair tax administration becomes paramount in achieving the increased revenue target (Directorate General of Taxes of Indonesia (DGT), 2020).

The commonly employed method for assessing the efficiency of various institutions is data envelopment analysis (DEA). This technique evaluates the efficiency of work units that utilize multiple inputs to achieve desired outcomes. DEA finds extensive application in measuring the performance of diverse entities, including banks, companies, governments, research institutions, and hospitals. It is considered a nonparametric estimation method for assessing the relative efficiency of these entities (Zhang, Xiao & Niu, 2022). Originally introduced by Charnes, Cooper & Rhodes (1978), DEA has become widely recognized as a modern and valuable tool for efficiency measurement (Rostamzadeh et al., 2021). In recent years, there has been a significant upsurge in publications concerning the theory and application of DEA (Emrouznejad & Yang, 2018). Several studies have employed DEA to evaluate the efficiency of tax agencies in various regions, such as Spain (González & Rubio, 2013), OECD countries (Alm & Duncan, 2014), Brazil (De Carvalho Couy, 2015), Taiwan (Huang, Yu & Huang, 2022), and African countries (ATAF, 2021). In the context of Indonesia, the efficiency of tax service offices has been examined in several instances, including the East Java Regional Office (Triantoro & Subroto, 2016), all tax service offices in 2011 (Suyanto & Saksono, 2013), and 2012 (Fadhila, 2014).

In the context of DEA, an important concern is the need for homogeneity among the decision-making units (DMUs) being measured (Omrani, Shafaat & Emrouznejad, 2018), However, existing DMUs often lack this homogeneity (Razavi Hajiagha, Hashemi & Amoozad Mahdiraji, 2016), necessitating a method to maintain uniformity within the DMU population. To address the issue of DMU heterogeneity, researchers have explored clustering as a solution. Some studies have combined machine learning and DEA methods to evaluate the efficiency of hospitals (Omrani, Shafaat & Emrouznejad, 2018) and banks (Razavi Hajiagha, Hashemi & Amoozad Mahdiraji, 2016). These studies utilize machine learning algorithms to cluster DMUs into homogeneous groups based on input and output variables.

The researchers also integrated DEA with machine learning regression algorithms and have applied it in various domains, including manufacturing companies (Zhu, Zhu & Emrouznejad, 2021), carbon emissions (Zhang, Xiao & Niu, 2022), and bank efficiency in China (Dalvand et al., 2014). In such studies, machine learning algorithms are used to predict the efficiency value of new dynamic data after DEA generates the static efficiency value. The static efficiency score serves as training and testing data for the regression machine learning algorithm.

To address the heterogeneity problem and the limitation of DEA in measuring dynamic efficiency against new data, researchers have undertaken studies that integrate machine learning and DEA methodologies. Some of these studies focused on clustering, measuring, and predicting the efficiency value of poultry farming companies in Iran (Rahimi & Behmanesh, 2012) and the performance of companies in the Tehran stock market (Rezaee, Jozmaleki & Valipour, 2018). In these investigations, machine learning algorithms were employed to initially cluster the DMUs, then DEA was used to calculate static efficiency values, and regression techniques were applied to predict dynamic efficiency values for new data. However, it is important to note that these studies clustered all input and output variables together.

This article aims to measure the efficiency value of all tax service offices in Indonesia in 2022 via DEA and machine learning through three stages. The first stage uses machine learning to cluster DMUs to overcome heterogeneity issues. In this stage, clustering has been performed only on input variables to objectively quantify the efficiency value. The second stage is measuring the static efficiency value via the DEA method. The third stage involves predicting the efficiency value based on the combination of input and output variables when designing a new office using regression algorithm.

Methods

This research employs a comprehensive methodology consisting of three approaches to analyze the efficiency of tax offices: clustering techniques, DEA, and regression modeling. The techniques were implemented in Python 3.11, using libraries such as gurobipy for DEA analysis, scikit-learn for modeling, and common libraries like pandas, numpy, and matplotlib. Clustering: This technique is used to group data into more homogeneous clusters, effectively addressing the issue of heterogeneity within the dataset. We employ various clustering algorithms in machine learning, including fuzzy c-means (FCM), density based spatial clustering of applications with noise (DBSCAN), k-medoids, and ordering points to identify the clustering structure (OPTICS). To evaluate the quality of the clustering results, we utilize the Davies-Bouldin Index (DBI) and Silhouette score, which provide insights into the separation and cohesion of the clusters formed.

DEA: It is used to measure the relative efficiency of business units or organizations, enabling the assessment of performance in comparison to other entities. In our analysis, we apply the DEA with the variable returns to scale (VRS) approach, using both input-oriented and output-oriented models.

Regression: This technique is employed as a predictive tool to forecast dynamic efficiency values for planning purposes, particularly in scenarios where input and output data for new DMUs are unavailable for direct DEA application. This approach is particularly useful in determining the distribution of input and output variables when forming new office in order to achieve efficiency. In this analysis, we utilized multilayer perceptron regressor (MLPR), support vector regressor (SVR), Random Forest regressor (RFR), and gradient boosting regressor (GBR). To prevent overfitting, we applied k-fold cross-validation, specifically using five folds. Additionally, hyperparameter tuning was conducted using a genetic algorithm. The model’s performance was evaluated using the mean squared error (MSE), providing insight into the accuracy of the predictions. Additionally, standard deviation was used as a metric to further evaluate the stability and reliability of the model predictions.

The combination of these three approaches is expected to yield a more comprehensive understanding and accurate outcomes in the analysis. This is based on our previous experience combining several approaches into a unified framework (Murakami et al., 2012).

Machine learning enables computers to make decisions based on data without being explicitly programmed for every possible scenario. However, the learning process itself relies on a well-defined and structured programming framework to develop models and extract insights from data (Samuel, 2000). By utilizing machine learning, data interpretation becomes more manageable, especially with the large volumes of data available today. Numerous industries have embraced machine learning to extract meaningful information and knowledge relevant to their activities. Machine learning relies on a diverse set of algorithms to solve various data-related problems. Data scientists understand that there is no one-size-fits-all algorithm for problem-solving. The choice of algorithm depends on factors such as the specific problem at hand, the number of variables involved, the most suitable model type, and other relevant considerations. This adaptability allows machine learning to be applied effectively to a wide range of tasks and industries.

Clustering is a fundamental technique in machine learning and data analysis that aims to group a set of objects in such a way that objects in the same group, or cluster, are more similar to each other than to those in other groups. This technique is particularly useful in exploratory data analysis, allowing researchers to discover patterns, structures, and relationships within large datasets. Clustering algorithms can be categorized into several types, including partitioning methods like k-means, hierarchical methods, density-based methods such as DBSCAN and OPTICS, and soft clustering methods like FCM. These algorithms facilitate tasks such as customer segmentation, image analysis, and anomaly detection, providing valuable insights across various domains. The effectiveness of clustering often depends on the choice of algorithm, the quality of the data, and the definition of similarity (Jain, 2010).

FCM is a non-hierarchical clustering technique within fuzzy clustering methods. It was initially introduced by Dunn in 1973 and further developed by Bezdek in 1981 (Rezaee, Jozmaleki & Valipour, 2018). While similar to the k-means method, FCM incorporates the concept of fuzzy theory to enhance clustering outcomes (Ye & Jin, 2016). In the FCM approach, fuzzy memberships are used, which provide membership degrees for each data point to multiple clusters (Nayak, Naik & Behera, 2015). The process of FCM data clustering begins with an initial estimation of the cluster center. Each data point is then assigned a certain degree of membership to each cluster. The algorithm iteratively updates the cluster centers and reassigns data points to the cluster they are closest to. This iterative process aims to minimize the objective function of the FCM method. The objective function of the FCM method can be represented by the following equation (Bezdek, Ehrlich & Full, 1984):

Jm=∑i=1n∑j=1cuijm||xi−vj||2

where Jm represents the objective function to be minimized, uij denotes the degree of membership of the data point xi in cluster j, while m is the fuzziness exponent that controls the fuzziness of the membership values, with m>1, xi represents the i-th data point, and vj is the centroid of cluster j, ||xi−vj||2 represents the squared Euclidean distance between data point xi and centroid vj. The summations are carried out over all n data points and c clusters.

K-medoids is a clustering algorithm designed to partition a dataset into a specified number of clusters using medoids as the cluster centers. A medoid is the most representative data point within a cluster, distinguishing it from the k-means algorithm, which uses the centroid (the average of all points in the cluster). The k-medoids algorithm begins by randomly selecting a set of medoids and then clusters the data points based on their proximity to these medoids. It optimizes the clustering by minimizing the total dissimilarity between the data points and their corresponding medoids. One of the key advantages of k-medoids over k-means is its robustness to outliers; medoids are less influenced by extreme values than centroids. This algorithm is particularly effective for clustering smaller datasets and demonstrates greater resilience to noise in the data (Kaufman, 1990).

DBSCAN is presented as a clustering technique that groups data points based on density. The algorithm defines clusters as areas where data points are densely packed, separated by regions of lower density. DBSCAN categorizes points into core points, which have enough neighboring points within a specified distance (Eps); border points, which are close to core points but lack sufficient neighbors to be considered core themselves; and noise points, which do not belong to any cluster. This approach allows DBSCAN to effectively discover clusters of arbitrary shapes, manage noisy data, and eliminate the need to specify the number of clusters in advance, making it highly suitable for large-scale spatial data (Ester et al., 1996).

OPTICS is also a density-based algorithm but is designed to address some of the limitations of DBSCAN. While DBSCAN produces distinct clusters, OPTICS generates an ordering of points that reflects the cluster structure and data density. Using the same parameters as DBSCAN, OPTICS retains information about data density and can differentiate between clusters that have varying densities. This allows OPTICS to build a hierarchy of clusters and perform better in managing data with varying densities, providing users with the flexibility to determine clusters based on different levels of density (Ester et al., 1996).

DEA is nonparametric mathematical programming that is essentially advanced linear programming based on a frontier estimation approach (Coelli, 1996). Nonparametric refers to statistical methods that do not require any parameter assumptions for the population being tested (Wolfowitz, 1949). DEA, introduced by Charnes, Cooper & Rhodes (1978), is a method of efficiency analysis used to assess how effectively a business unit or organization utilizes its available inputs to achieve the highest possible output. By comparing the use of inputs and relative outputs among different units, DEA generates a relative efficiency value, allowing for a comparison of the performance of various business units or organizations. The research steps of the DEA method involve identifying the DMUs or units to be observed, along with their respective inputs and outputs. Efficiency is then calculated for each DMU, providing the input and output targets required to achieve optimal performance (Indrawati, 2009). Initially developed to evaluate non-profit and government organizations, DEA was later applied to assess the performance of service operations in various private companies (Sherman & Zhu, 2013).

The selection of appropriate input and output variables in DEA is critical, as using irrelevant variables can bias the analysis and lead to inaccurate conclusions. In this study, input variables were selected based on a thorough review of previous research on DEA’s application in measuring the efficiency of tax service offices. This approach ensures alignment with the operational framework of tax offices in Indonesia. Additional input indicators, such as those proposed by Milosavljević, Radovanović & Delibašić (2023), could be considered for future analyses. DEA models vary in their treatment of variable returns to scale. The two most common models, DEA-CCR (Charnes, Cooper, Rhodes) and DEA-BCC (Banker, Charnes, Cooper), offer different perspectives on efficiency evaluation depending on the specific characteristics of the analyzed units (Charnes, Cooper & Rhodes, 1978; Banker, Charnes & Cooper, 1984).

The DEA-CCR model can be customized based on output or input, and the choice of this model depends on the characteristics of DMU in the production frontier. Input-oriented models minimize the inputs for a given level of outputs, whereas output-oriented models maximize the production of outputs for a given level of inputs. Suppose there are n DMUs, and each DMUj(j=1,2...n) produces s output yrj(r=1,....s) using m inputs xij(i=1,...m); then DEA-CCR uses the following equation to evaluate the efficiency of the DMU:

maxθsubjectto∑j=1nλjyrj≥θyr0,r=1,2,...,s∑j=1nλjxij≤xi0,i=1,2,...,mλj≥0,j=1,2,...,n

where θ represents the efficiency score to be maximized. The term λj refers to the weight assigned to DMU j. The r-th output for DMU j is denoted by yrj, while xij stands for the i-th input of DMU j. The values yr0 and xi0 represent the outputs and inputs of the DMU under evaluation, labeled as DMU 0. The total number of DMUs is n, the number of inputs is m, and s is the number of outputs.

In practical applications, the original nonlinear equation of the DEA-BCC model, with infinite optimal solutions, needs to be transformed into a suitable pairwise linear programming model. This conversion ensures that the efficiency evaluation can be effectively implemented. The transformed equation takes the following form (Zhang, Xiao & Niu, 2022):

minθsubjectto∑j=1nλjxij≤θxik∑j=1nλjyrj≥yrk0<θ≤1;λ≥0;i=1,2,…,m;r=1,2,…,q;j=1,2,…,n;k=1,2,…,s

where θ represents the efficiency score of the DMU under evaluation and is to be minimized, λj denotes the non-negative weight assigned to DMU j in the linear combination, xij refers to the input of DMU j for input category i, while xik is the input of the DMU being evaluated. Similarly, yrj represents the output of DMU j for output category r, and yrk is the output of the DMU under evaluation. Here, k represents the DMU being evaluated, and j denotes the other DMUs used for comparison.

The DEA-BCC model is a variant of DEA that assumes variable returns to scale. This means it assumes that the DMU is operating at an optimal scale. However, DEA-BCC also incorporates the notion of variable returns to scale, implying that changes in inputs may not result in a proportional change in outputs (Banker, Charnes & Cooper, 1984). The DEA-BCC model is particularly suitable for measuring efficiency in the public sector, where operations may not always be at an optimal scale (Kalb, 2010). It allows for a more realistic assessment of efficiency in such contexts. The key distinction between the DEA-BCC and DEA-CCR models lies in the constraints imposed on each weight λ in the equation of the DEA-CCR model. These constraints are modified in the DEA-BCC model, resulting in the following equation (Banker et al., 1989):

minθsubjectto∑j=1nλjxij≤θxik∑j=1nλjyrj≥yrk∑knλk=10<θ≤1;λ≥0;i=1,2,…,m;r=1,2,…,q;j=1,2,…,n;k=1,2,…,s

where θ represents the efficiency score of the DMU under evaluation and is to be minimized. λj represents the non-negative weights assigned to each DMU j in the linear combination. These weights determine how much each DMU contributes to the combination. xij denotes the input i used by DMU j, while xik represents the input i used by the DMU being evaluated (DMU k). Similarly, yrj represents the output r produced by DMU j, and yrk is the output r produced by the DMU under evaluation. Here, k refers to the DMU being evaluated, and j refers to the other DMUs used for comparison in the linear combination.

In the DEA-BCC model, the efficiency values obtained from the input-oriented and output-oriented approaches are different. Consider the point C, as illustrated in Fig. 1. To calculate the input-oriented efficiency value at point C, we divide the distance QC1 by the distance QC. On the contrary, to calculate the output-oriented efficiency value at point C, we divide the distance NC by the distance NC2.

Figure 1 Illustration of input-oriented and output-oriented efficiency models in DEA.

The input-oriented approach in DEA-BCC focuses on efficiently using inputs to produce predetermined outputs. In input-oriented DEA-BCC, DMUs are considered units that minimize the use of inputs to produce predetermined outputs. In the input-oriented DEA-BCC model, the efficiency frontier construction technique is employed to evaluate the relative efficiency level of each DMU in utilizing their inputs. DMUs located on the efficiency frontier in input-oriented DEA-BCC are considered efficient in minimizing the usage of inputs to produce the specified outputs. These efficient DMUs serve as benchmarks for other units to strive for in terms of input utilization efficiency. The input-oriented DEA-BCC formula can be represented by the following equation (Banker, Charnes & Cooper, 1984):

minθsubjectto∑j=1nλjxjk≤θ⋅xkk∑j=1nλjyji≥yki∑j=1nλj=10<θ≤1;λ≥0;i=1,2,…,m;r=1,2,…,q;j=1,2,…,n;k=1,2,…,s

where θ represents the input-oriented efficiency score of DMU k. The weights λj create a composite DMU from the inputs and outputs of other DMUs. The constraint ∑j=1nλjxjk≤θ⋅xkk ensures that the total input of the composite DMU does not exceed the scaled input of DMU k. The constraint ∑j=1nλjyji≥yki ensures that the output of the composite DMU is at least as large as the output of DMU k. The constraint ∑j=1nλj=1 ensures that the weights sum to 1, allowing for a proportional adjustment of inputs and outputs. Finally, λj≥0 ensures that all weights are non-negative.

The output-oriented DEA-BCC approach focuses on the output produced by DMUs using predetermined inputs. In output-oriented DEA-BCC, DMUs are considered units that use specific inputs to produce the most efficient output possible. Conversely, output-oriented DEA-BCC uses efficiency frontier construction techniques to determine the relative efficiency level of each DMUs in producing their outputs. Decision-making units on the efficiency frontier in output-oriented DEA-BCC are considered efficient in utilizing available inputs to produce the maximum output. The output-oriented DEA-BCC calculation can be represented in the following equation (Banker, Charnes & Cooper, 1984):

maxθsubjectto∑j=1nλjyji≥θ⋅yki∑j=1nλjxjk≤xkk∑j=1nλj=10<θ≤1;λ≥0;i=1,2,…,m;r=1,2,…,q;j=1,2,…,n;k=1,2,…,s

where θ represents the output-oriented efficiency score of DMU k. The weights λj combine the outputs and inputs from other DMUs to create a virtual DMU for comparison. The constraint ∑j=1nλjyji≥θ⋅yki ensures that the combined output of the virtual DMU is at least equal to the output of DMU k, scaled by θ. The constraint ∑j=1nλjxjk≤xkk ensures that the input used by the virtual DMU does not exceed the input used by DMU k. The constraint ∑j=1nλj=1 ensures that the weights sum to 1, allowing for scaling adjustments. Finally, λj≥0 ensures that the weights are non-negative.

Regression in the field of machine learning is a supervised learning technique used to predict continuous (numeric) values of a target variable based on one or more input features. The main goal of regression is to build a model that learns the relationship between inputs and outputs from training data, enabling the model to make accurate predictions on previously unseen (testing) data. This technique is widely utilized in fields such as economics, finance, biology, and social sciences to make informed predictions and decisions (Montgomery, Peck & Vining, 2021).

Support vector regressor (SVR) is a machine learning algorithm derived from support vector machines (SVM), primarily used for predicting continuous values in regression tasks. It finds a function that deviates from actual observed values by no more than a specified threshold (epsilon), aiming to minimize error within this margin while maintaining generalization for unseen data. SVR utilizes kernel functions to address non-linear relationships and establish complex decision boundaries, with common kernels including linear, polynomial, and radial basis function (RBF). The algorithm is effective in high-dimensional spaces and robust against overfitting, though it can be sensitive to parameter choices, such as regularization and kernel type (Smola & Schölkopf, 2004).

Random Forest regressor (RFR) is an ensemble learning method that operates by constructing multiple decision trees during training and outputting the average prediction from these trees for regression tasks. It combines the predictions of numerous trees, which helps mitigate the overfitting often seen in individual decision trees and enhances overall model accuracy. The algorithm operates by randomly sampling subsets of data and features, ensuring diversity among the trees, which contributes to its robustness and effectiveness in capturing complex patterns in the data. One of its key advantages is the ability to handle large datasets with high dimensionality while providing insights into feature importance (Breiman, 2001).

Gradient boosting regressor (GBR) is an ensemble learning technique that enhances predictive performance by sequentially combining multiple weak learners, typically decision trees. The method focuses on correcting the errors made by previous trees, with each new tree added to the ensemble aimed at minimizing the residuals of the combined model from earlier iterations. This optimization is achieved through gradient descent on a specified loss function, enabling the model to capture complex relationships and feature interactions. While gradient boosting is effective for various regression tasks, it is sensitive to overfitting, particularly with a high number of trees, requiring careful tuning of parameters like learning rate and tree depth (Friedman, 2001).

Multilayer perceptron (MLP) networks are among the most popular artificial neural networks used in various scientific fields, particularly in forecasting and prediction. MLP networks consist of an input layer, one or more hidden layers (collectively referred to as the hidden part of the network), and an output layer, as illustrated in Fig. 2. The input layer receives a vector of data or patterns. The hidden part of the network contains one or more layers that process inputs from the preceding layer, assign weights, and pass the weighted sums through an activation function. In this study, we use the ReLU activation function because it only produces positive values, which aligns well with the nature of DEA that only generates positive outputs. The output layer takes the outputs from the last hidden layer, assigns weights, and potentially passes them through an output activation function to produce a target value.

Figure 2 Illustration of multilayer perceptron.

K-fold cross-validation is a commonly used method for evaluating machine learning models, dividing the dataset into k equal-sized folds where k-1 folds are used for training and the remaining fold is used for validation. This process is repeated k times, allowing each fold to serve as a validation set once, and the final performance is averaged across all iterations. K-fold cross-validation helps prevent overfitting by validating the model on multiple partitions of the data. Variants like stratified k-fold are used for handling imbalanced datasets to ensure consistent class distribution across the folds. The method was influenced by the development of resampling techniques, particularly the work of Efron & Tibshirani (1993) on the bootstrap, and gained wider recognition in machine learning following Kohavi’s study on accuracy estimation and model selection (Kohavi, 1995).

Genetic algorithm (GA) belongs to the class of evolutionary algorithms and is inspired by natural selection cycles (Mitchell, 1998). It is a powerful optimization algorithm from the traditional heuristic family, well-suited for handling solutions trapped in local minima. In machine learning optimization, conventional algorithms like gradient descent and grid search may stop at a suboptimal solution due to the risk of getting stuck in a local minimum. However, GA can surpass these local minima and achieve globally better solutions. GA achieves this by employing selection, crossover, and mutation mechanisms to maintain variation within the population and avoid being trapped at a local minimum. It is particularly effective for problems requiring optimization within a countable system (Lambora, Gupta & Chopra, 2019). In the implementation of GA, a population of candidate solutions evolves iteratively toward a better solution. Each member of the population has a set of characteristics that can change and undergo mutation. The process begins with the initial formation of a population comprising randomly generated individuals. Each iteration, or generation, in the GA, involves calculating the fitness of each member. Fitness usually represents the value of the objective function specific to the problem being solved. Members with higher fitness are then selected from the current population, and their characteristics are combined through crossover to produce offspring with inherited characteristics (Gajić et al., 2020). The GA used in this study can be explained with the following algorithm: Initialize the maximum number of generations and population size n

Generate an initial random population of n solutions

While the number of generations has not reached the maximum: Evaluate the fitness function f(x) for each solution in the population

Create offspring until the desired number is reached: – Select two parent solutions from the population using the roulette wheel selection method.

– Apply the crossover operator to the selected parents with probability p, producing two offspring.

– Apply the mutation operator to the offspring with a probability equal to the mutation rate.

Replace the current population with the newly generated offspring.

Terminate when the maximum number of generations is reached or other stopping criteria are met.

There are several methods used to calculate network error, one of which is MSE. MSE measures the average of the squared difference between the predicted value and actual value. The smaller the MSE value, the better the model predicts the data. MSE can be used if there are outliers in the observed data (Chicco, Warrens & Jurman, 2021). The MSE calculation can be represented in the following equation:

MSE=1n∑i=1n(yi−y^i)2

where n is number of data points, yi is actual value, and y^i is predicted value.

In the context of k-fold cross-validation, standard deviation measures the variability of the model’s performance metrics (e.g., MSE) across the k folds. A low standard deviation indicates that the model performs consistently across all folds, suggesting robustness and stability, while a high standard deviation may highlight inconsistencies in model behavior across different subsets of the data. In this study, we used Python’s scikit-learn library to calculate the standard deviation of the performance metrics across the folds, providing insights into the model’s reliability.

Related studies

Previous studies on efficiency measurement using DEA and machine learning have been utilized as references in this current study. These studies can be categorized into four main groups based on their focus areas: DEA used in taxation, machine learning for DMU clustering in DEA, machine learning for dynamic efficiency prediction, machine learning in DEA for clustering, and dynamic efficiency prediction.

DEA on tax service office field

Previous studies using DEA in the field of taxation are presented in Table 1. None of these studies in the field of taxation use the clustering method. Therefore, heterogeneity is probable. In addition, it has no regression method of measuring the dynamic efficiency value of new data.

Table 1 DEA on tax service office field.

Research works	Contributions	
González & Rubio (2013)	Measuring the efficiency value of tax administration performance in Spain using DEA, without clustering DMUs and predicting dynamic efficiency for new data.	
Alm & Duncan (2014)	Determining the relative efficiency of tax agents in OECD member countries using DEA, without DMU clustering and dynamic efficiency prediction for new data.	
De Carvalho Couy (2015)	Measuring the efficiency in tax audit performance at the Brazilian tax authority using DEA, without dynamic efficiency prediction for new data and grouping DMUs to cluster.	
Triantoro & Subroto (2016)	Measuring the efficiency performance of tax service offices using DEA without clustering and predicting dynamic efficiency for new data.	
Huang et al. (2017)	Measuring efficiency of tax collection and tax management in Taiwan’s local tax service offices using DEA without clustering and predicting dynamic efficiency for new data.	
Suyanto & Saksono (2013)	Analyzing the efficiency of tax service offices in Indonesia using DEA without clustering and predicting dynamic efficiency for new data.	
ATAF (2021)	Evaluation of tax administration efficiency of African tax administration forum member countries using DEA without clustering and predicting dynamic efficiency for new data.	

DEA and clustering

Previous studies that use machine learning on DEA to cluster DMUs are presented in Table 2. All studies using machine learning to cluster DMUs were conducted on input and output variables. In addition, these previous studies have not employed any regression method to predict the dynamic efficiency value of new data.

Table 2 DEA and clustering machine learning researches.

Research works	Contributions	
Razavi Hajiagha, Hashemi & Amoozad Mahdiraji (2016)	Integrating FCM and DEA to mitigate DMU heterogeneity in banks. Clustering is performed on the input and output variables, It does not involve regression prediction using machine learning.	
Omrani, Shafaat & Emrouznejad (2018)	Integrating fuzzy clustering and DEA to find efficiency in hospitals in Iran. The input and output variables are clustered with no regression utilized.	

DEA and prediction

Previous studies using machine learning on DEA to predict the dynamic efficiency of new data without clustering DMUs are shown in Table 3. All studies that use machine learning to predict the new data do not cluster the DMUs. Hence, DMU heterogeneity is feasible, resulting in efficiency values to be less objective.

Table 3 DEA and machine learning prediction researches.

Research works	Contributions	
Dalvand et al. (2014)	Integrating C4.5 classification algorithm and DEA to predict static and dynamic efficiency values for 200 bank branches in Iran. Machine learning is solely used to predict efficiency values for new data, not to cluster DMUs to obtain DMU homogeneity.	
Appiahene, Missah & Najim (2020)	Combining DEA with three machine learning approaches to evaluate the efficiency and per formance of banks using 444 bank branches in Ghana. Only efficiency values for new data are predicted using machine learning and DMUs are not clustered to achieve DMU homogeneity.	
Zhu, Zhu & Emrouznejad (2021)	Combining DEA and machine learning to measure and predict the efficiency values of manufacturing companies in China. Instead of clustering DMUs to achieve DMU homogeneity, machine learning is only employed to predict efficiency values for new data.	
Zhang, Xiao & Niu (2022)	The paper specifically uses the case of China’s regional carbon emission performance prediction to demonstrate the effectiveness of the proposed integrated model of DEA and machine learning. No DMUs are clustered to achieve DMU homogeneity; instead, machine learning is utilized to predict efficiency values for new data.	

DEA and clustering-prediction

Previous studies that have applied machine learning methods in DEA to cluster DMUs and predict efficiency on new data are shown in Table 4. The results of these studies show that the DMU clustering stage is conducted on the input and output variables. This step may lead to a potential lack of objectivity in the efficiency assessment of the clusters formed.

Table 4 DEA and clustering-prediction machine learning researches.

Research works	Contributions	
Rahimi & Behmanesh (2012)	Combining DEA and data mining techniques, such as artificial neural network, and decision tree, to predict the efficiency of poultry companies in Iran by clustering DMUs on input and ouput variables.	
Rezaee, Jozmaleki & Valipour (2018)	Integrating FCM, DEA, and machine learning to measure the performance of companies in the stock exchange by clustering DMU input and output variable.	

Approach

This study utilizes an integrated framework by combining machine learning and DEA to measure and predict the performance efficiency of tax service offices in Indonesia as the measured DMU. This framework is divided into three processes: Clustering, which uses machine learning algorithms to achieve DMU homogeneity; DEA, for measuring static efficiency; and Regression, for measuring dynamic efficiency with new data, as shown in Fig. 3.

Figure 3 Flowchart illustrating the outlined approach.

Before carrying out these three stages, data preprocessing is carried out so that the data can be processed further. The data used in this study was sourced from the Directorate General of Taxes in Indonesia. Subsequently, the dataset undergoes a data understanding process, which aims to ascertain its suitability for immediate consumption or if specific actions are required before further processing. This analysis includes assessing the data structure, identifying variables with negative values that are not suitable for the DEA model, and detecting potential data outliers. In addition, this stage also includes the separation of input and output variables. The selection of input variables is based on analysis from previous research, which has been adapted to the operations of tax service offices in Indonesia and verified by the authorities.

Input variables consist of: Vin1: Number of corporate taxpayers

Vin2: Number of treasury taxpayers

Vin3: Number of individual taxpayers

Vin4: Number of non-employee taxpayers

Vin5: Number of tax auditors

Vin6: Number of account representatives

Vin7: Budget realization amount

Output variables consist of: Vout1: Compliance rate of annual tax return submission

Vout2: Percentage of revenue achievement

Vout3: Percentage of revenue growth achievement

Vout4: Number of issued tax advisories

Vout5: Number of paid tax advisories

Vout6: Number of completed tax audits

Clustering

The first stage involves the clustering process to categorize DMUs into several clusters for enhanced homogeneity. Machine learning algorithms like k-medoids, FCM, DBSCAN, and OPTICS are employed for clustering, and their effectiveness is assessed using the silhouette value and Davies Bouldien Index (DBI). Higher silhouette values indicate more accurate clustering, while lower DBI values signify better cluster quality. The selection of these clustering techniques is based on previous experience using the basic k-means algorithm for document clustering (Usino et al., 2019).

Based on preliminary experiments with ideal data, specifically data that contains four combinations of low and high input-output variables, as shown in Table A1, it is proposed to cluster the input variables alone. This approach yields a more objective efficiency value compared to clustering both input and output variables. The experiment’s results, presented in Table 5, show that clustering inputs and outputs produced four clusters, maximizing efficiency results due to the relative nature of DEA. In contrast, clustering only the input variables resulted in two clusters, with maximum and minimum efficiency values.

Table 5 Clustering simulation and efficiency results.

Input and output combination	Input and output clustering	Input clustering only	
Input	Ouput	Cluster	Efficiency	Cluster	Efficiency	
Low	Low	C1	Max	C1	Min	
Low	High	C2	Max	C1	Max	
High	Low	C3	Max	C2	Min	
High	High	C4	Max	C2	Max	

This lack of objectivity can be observed in the combination of low input and output variables, leading to the maximum value in the input-output clustering. This situation is less objective compared to the combination of low input and high output variables, both of which produce maximum efficiency values. This situation is different from the clustering of input variables only. The combination of the low input and output variables produces a minimum efficiency value, and the combination of the low input and high output variables produces a maximum efficiency value. This phenomenon is also observed in the combination of high input and low output variables compared with the combination of high input and output variables. The clustering of input variables is more objective than the clustering of input and output variables.

DEA process

The second stage of the process involves using DEA to calculate static efficiency values for each cluster formed in the previous clustering stage. DEA determines these static efficiency values based on historical data. Efficiency values are represented on a scale from 0 to 1, with 1 indicating maximum efficiency. DEA compares the utilization of exploiting inputs and the production of outputs among the units. Units with higher efficiency values are considered more efficient in resource utilization. The analysis helps identify units with the potential to improve their efficiency by adopting best practices from the most efficient units. At this stage, the results obtained from clustering the input variables are combined with their corresponding output variables, creating clusters consisting of both types of variables. These combined clusters are then subjected to DEA analysis to determine their static efficiency values. The DEA process utilizes the BCC method, assuming that all DMUs in the cluster have not yet reached the optimum performance level. This method employs input-oriented and output-oriented approaches, with a focus on identifying the optimal combination of inputs to produce a given output. The main objective is to measure the relative efficiency of each DMU in achieving optimal results while utilizing available resources. Using the BCC method, the DEA stage of the process can furnish information on the static efficiency level of each pre-formed cluster, taking into account the relevant input and output variables. This approach aids in comprehending the efficiency of resources for each cluster and highlights areas where improvements can be made to achieve higher levels of efficiency.

Regression process

The final stage is the regression process, which predicts dynamic efficiency values based on new data that does not exist in historical records. This stage is applied to data for offices that have not yet been established, as they have neither inputs nor outputs. It serves as a simulation tool to determine the optimal combination of input and output variables when setting up a new tax service office to achieve maximum efficiency. Additionally, this regression can also evaluate existing tax service offices, identifying opportunities to improve efficiency by adjusting the values of their input and output variables.

To achieve this, several machine learning regression algorithms are employed, including GBR, MLPR, SVR, and RFR. Prior to applying these regression algorithms, k-fold cross-validation is employed with K = 5 to assess model performance and mitigate the risk of overfitting. This technique involves dividing the training data into five subsets, training the model on four of these subsets, and validating it on the remaining subset. This process is repeated five times, ensuring that each subset is used for validation once, which helps provide a more robust estimate of model performance. The regression process utilizes the static efficiency results obtained from the DEA stage as training data. The independent variables (x) consist of the input and output variables, while the dependent variable (y) is the static efficiency value generated in the DEA process.

Initially, the regressor models are created, and their performance is then optimized using the GA. The GA is an optimization method inspired by natural evolution principles. It is used in this context to find the best configuration of model parameters for each regressor. These parameters may include the number of trees, the depth of the tree (max depth), the number of neurons in the hidden layer (for MLPR), and the learning rate. The GA will iteratively experiment with various parameter combinations, evaluate each model’s performance, and select the best configuration based on the objective function. In this study, the MSE value is used as the objective function for model evaluation. The aim is to minimize the MSE and create the most accurate regression model for predicting dynamic efficiency values. Additionally, standard deviation is monitored to assess the variability of the predictions and ensure the robustness of the final model.

Result and discussions

This study uses data derived from population data of tax service offices in Indonesia, with samples presented in Table A2. Before conducting the analysis, it is necessary to merge the data from both sources and adjust the format accordingly. The dataset comprises 352 rows and 14 columns. The DMU column contains three-digit identity codes of the tax service offices in a masked form. The details of 14 columns can be seen in Table A3. The columns comprise one “DMU” column serving as the identity and primary key, seven columns of input variables, and six columns of output variables.

Before proceeding with data processing, duplicate data detection is performed using methods such as edit distance, Jacobson, and cosine similarity. Fortunately, no duplicate data is found in the dataset, eliminating the need for duplicate data removal. Null or empty data detection is also conducted, and it is concluded that there are no null data entries in the dataset. As a result, no further steps are required for null data handling.

For the purpose of visualizing the data distribution, the principal component analysis (PCA) method is employed. PCA transforms the data into a two-dimensional representation, allowing for easier visualization and understanding. The results are displayed in Fig. 4, providing insights into the patterns and relationships between variables, aiding in comprehending the overall structure and distribution of the data. Notably, several outlier data points are identified, located far away from other data clusters. However, further analysis is required to confirm the presence of outliers at a later stage.

Figure 4 Data distribution visualization using a two-dimensional scatter plot in PCA.

The skewness analysis reveals that most variables in the dataset exhibit positive values, indicating a rightward skew in their data distributions. However, one variable, “vin6,” displays a negative skewness value, suggesting a left-skewed data distribution for this particular variable. A negative skewness value means that the tail of the data distribution tends to be longer on the left side of its center value. This finding highlights that the ”vin6” variable’s distribution asymmetry differs from the other variables. Therefore, when analyzing and interpreting the data, special attention should be given to the “vin6” variable due to its distinct distribution characteristics. Complete results of the skewness values can be found in Table A5.

To ensure comparability during data analysis and modification, the normalization stage is performed using various methods, including z-score scaler, min-max scaler, and log transformation. Tables A6, A7, and A8 present the results of the normalization process. Among these methods, only the min-max scaler results in all positive values. Since DEA requires all data to be positive to achieve more robust efficiency results, the min-max scaler method is chosen for further processing (Wei & Wang, 2017).

Additionally, in the data processing stage, outlier detection is carried out to identify any outlier data. The boxplot method is used for this purpose, and the results indicate the presence of outliers in each input variable. Outliers represent significant extreme values within the data, and their identification is crucial as they can influence the selection of clustering and regression algorithms. The findings of the outlier detection process are depicted in Fig. 5.

Figure 5 Outlier detection in boxplot chart.

Various clustering methods are used to group data into similar or homogeneous clusters. Because there were outliers in each input variable to be clustered, we used four clustering methods resistant to data outliers: k-medoids, FCM, DBSCAN, and OPTICS. Based on the experiment with five clusters from these four methods, K-Medoids and FCM provided the best silhouette score. The results of the silhouette score calculation for all the methods are shown in Table 6 below.

Table 6 Silhouette score for five clusters.

	K-MEDOIDS	OPTICS	FCM	DBSCAN	
Silhouette score	0.197174	0.094126	0.235802	0.061974	

Based on the initial five clusters’ results, the k-medoids and FCM algorithms were repeatedly tested to obtain the optimal number of clusters. The best results obtained in k-medoids are two clusters with a silhouette value of 0.265 and DBI 1.388, whereas the best number of clusters obtained in FCM is three clusters with a silhouette value of 0.304 and DBI 1.119. The test results of the number of clusters and silhouette value can be observed in Table 7 and Fig. 6

Table 7 Silhouette and DBI score for k-medoids and FCM.

Number of cluster	Silhouette score	Davies bouldin index	
	FCM	K-Medoids	FCM	K-Medoids	
2	0.270	0.265	1.382	1.388	
3	0.304	0.132	1.119	1.897	
4	0.249	0.091	1.308	1.852	
5	0.236	0.197	1.341	1.471	
6	0.107	0.156	2.335	1.570	
7	0.074	0.164	2.134	1.531	

Figure 6 (A) Best silhouette score for k-medoids clustering. (B) Best silhouette score for FCM clustering.

The clustering outcomes, featuring the most favorable silhouette scores from both the k-medoids and FCM techniques, can be effectively visualized via a scatter plot, offering valuable insights into cluster memberships. Figure 7 exhibits this scatter plot, presenting the clustering pattern in two dimensions through PCA. PCA serves to diminish the data’s high dimensionality, thereby simplifying the analysis and comprehension of intricate data. The plot illustrates a well-defined division of cluster members, with no instances of cluster overlap. The analysis leads to the conclusion that the FCM clustering algorithm, employing three clusters with a silhouette score of 0.304 and DBI of 1.119, outperforms the k-medoids algorithm, which employs two clusters and achieves a silhouette score of 1.265 and DBI of 1.388. A comprehensive comparison of these results is available in Table 7. Based on these findings, the FCM algorithm with three clusters was chosen for further investigation. The respective cluster and centroid data are visually presented in the two-dimensional scatter plot graph in Fig. 8. Additional information, including the cluster membership and centroid details, can be found in Table A9. Furthermore, the specific cluster results are detailed in Table A10. The clustering results can serve as recommendations for stakeholders in classifying offices into categories such as small, medium, and large.

Figure 7 (A) Scatterplot results of clustering with the best silhouette score value for k-medoids clustering. (B) Scatterplot results of clustering with the best silhouette score value for FCM clustering.

Figure 8 FCM cluster and centroid visualization.

After identifying the optimal clusters, the subsequent stage involves conducting DEA modeling. The primary objective of DEA is to determine static efficiency values for each DMU within each cluster. Each member of the cluster undergoes a separate DEA analysis. The efficiency values are computed using both input-oriented DEA-BCC and output-oriented DEA-BCC methods. These methods help assess the relative efficiency of each DMU within its respective cluster, taking into account input and output measures to evaluate their performance.

The detailed outcomes of the input-oriented DEA-BCC method can be observed in Table A11. From the results, it can be deduced that within cluster C0, 48 DMUs have attained efficiency, while the remaining 17 DMUs have not. In this cluster, the efficiency level reaches 74% of the total 65 DMU members. For cluster C1, 89 DMUs have achieved efficiency, accounting for 60% of the total DMUs, while the remaining 60 DMUs are not efficient. In cluster C2, 89 DMUs have already attained efficiency, constituting 65% of the total DMU members, leaving 49 DMUs yet to achieve efficiency.

The detailed results of the output-oriented DEA-BCC method can be found in Table A12. Within cluster C0, 54 DMUs have reached efficiency, and the remaining 11 DMUs have not yet achieved efficiency. In this cluster, the efficiency level reaches 83% of the total 65 DMU members. For cluster C1, 89 DMUs have attained efficiency, accounting for 60% of the total DMUs, while the remaining 60 DMUs are not efficient. In cluster C2, 88 DMUs have already attained efficiency, representing 63% of the total DMU members, with 50 DMUs yet to achieve efficiency.

The results of input-oriented and output-oriented calculations are then summarized in Table 8. For cluster C0, 48 DMUs, or 74% of the total 65 DMU cluster members, are efficient in both input-oriented and output-oriented approaches. In cluster C1, 89 DMUs, equivalent to 60% of the total cluster members, are efficient in both input-oriented and output-oriented analyses. In cluster C2, 88 DMUs, representing 64% of the total C2 cluster members, are efficient in both input-oriented and output-oriented evaluations. Overall, 225 out of 352 DMUs, or approximately 64%, demonstrated efficiency in both input- and output-oriented assessments. This information is summarized in Table A13. Consequently, it can be concluded that the overall performance of tax service offices in Indonesia has attained efficiency, with an efficiency level of 64% of the total number of offices. These findings indicate a substantial improvement over the previous study by Suyanto & Saksono (2013), which classified only 61 out of 331 tax service offices, or approximately 18%, as efficient.

Table 8 Summary of DEA process result.

Cluster	Number of cluster members	Efficiency	Input oriented	Output oriented	Input and output oriented	Percentage	
C0	65	Efficient	48	54	48	74%	
		Not Efficient	17	11	17	26%	
C1	149	Efficient	89	89	89	60%	
		Not Efficient	60	60	60	40%	
C2	138	Efficient	89	88	88	64%	
		Not Efficient	49	50	50	36%	
TOTAL	352	Efficient	226	231	225	64%	
		Not Efficient	126	121	127	36%	
		Sum	352	352	352	100%	

The results of the model optimization using GA are shown in Table 9. The MLPR algorithm optimized with GA (GA-MLPR) achieved the smallest objective function value of 0.0035, with an execution time of 8 minutes and 13 s, and it converged on the 13th iteration. The best parameter configuration consists of five hidden layers, 73 units per layer, a ReLU activation function, and a learning rate of 0.006. The second best-performing algorithm is the genetic algorithm SVR (GA-SVR) with an objective function value of 0.0037, followed by the genetic algorithm RFR (GA-RFR) with an objective function value of 0.0051, and the genetic algorithm gradient boosting regressor (GA-GBR) with an objective function value of 0.0052. These results show a significant decrease in the MSE value for GA-MLPR, which decreased from 0.0144 to 0.0035, reflecting a substantial improvement of 75.75%, as seen in Table 10. We chose GA-MLPR as the best model based on its lowest MSE (0.0035), which indicates the highest predictive accuracy among all tested models. The multilayer perceptron’s ability to capture non-linear relationships combined with genetic algorithms, offers good flexibility and adaptability to data variations while controlling the risk of overfitting. Although the standard deviation (0.0821) is slightly higher than some other models, it still indicates adequate stability, making GA-MLPR a robust and reliable solution for regression needs. The visualization of objective function values for each iteration can be seen in Fig. 9.

Table 9 Model optimization results with genetic algorithm.

Algorithm	Objective Function	Best solution	Executed time	Iteration	
GA-MLPR	0.0035	layer = 5	8 m 13 s	13	
		unit = 73			
		activation function = ReLU			
		learning rate = 0.006			
GA-SVR	0.0037	C = 0.549	14.2 s	5	
		epsilon = 0.0159			
GA-RFR	0.0051	n_estimators = 71	14 m 17 s	12	
		max_depth =10			
		max_features = 6			
GA-GBR	0.0052	n_estimator = 187.950	3 m 30 s	12	
		max_depth = 2.246			
		learning_rate = 0.09			

Table 10 Results of MSE and standard deviation value reduction before and after the algorithm is optimized using genetic algorithm.

Optimized algorithm	MSE	Decrease percentage	Standard deviation	
MLPR	0.0144	75.75%	0.1016	
GA-MLPR	0.0035		0.0821	
SVR	0.0057	34.28%	0.0714	
GA-SVR	0.0037		0.0696	
RFR	0.0059	13.78%	0.0728	
GA-RFR	0.0051		0.0724	
GBR	0.0057	8.92%	0.0713	
GA-GBR	0.0052		0.0712	

Figure 9 (A) Genetic algorithm chart optimization for GA-GBR. (B) Genetic algorithm chart optimization for GA-MLPR. (C) Genetic algorithm chart optimization for GA-SVR. (D) Genetic algorithm chart optimization for GA-RFR.

Conclusions

This article presents an experimental approach to assess the efficiency of tax service offices in Indonesia using real dataset through three stages: clustering with k-medoids, OPTICs, DBScan, and FCM algorithms to group tax service offices as DMUs; static efficiency measurement using input-oriented and output-oriented DEA-BCC; and dynamic efficiency prediction using machine learning regression algorithms (gradient boosting regressor, multilayer perceptron regressor, support vector regressor, and Random Forest regressor) optimized with GA. The FCM algorithm, with a silhouette value of 1.304 and a DBI value of 1.119, outperformed other algorithms and produced three clusters of tax service offices.

In the DEA measurement, using the input-oriented DEA-BCC method, 226 tax service offices were found to be efficient DMUs, while using the output-oriented DEA-BCC method, there were 231 efficient DMUs. Overall, 225 out of 352 DMUs demonstrated efficiency in both input- and output-oriented calculations, representing 64% efficient DMUs of the Tax Service Office. These findings show a significant improvement over the previous study, in which only 61 out of 331 tax service offices, or about 18%, were classified as efficient.

The MLPR algorithm optimized with genetic algorithm (GA-MLPR) obtained optimal results with a parameter combination of 73 units per layer, five hidden layers, a ReLU activation function, and a learning rate of 0.006. It achieved an objective function value of 0.0035 during the 13th iteration, significantly reducing the MSE value by approximately 75.75% from 0.0144 to 0.0035.

The findings of this study can serve as a reference for stakeholders to categorize tax offices into small, medium, and large categories based on the clustering results. The DEA process that identifies efficient offices can serve as a benchmark for the efficiency levels that other offices should aim to achieve. Additionally, stakeholders can propose these efficient offices for incentives as a form of reward, which is expected to motivate performance improvement across the tax service sector.

For future research, it is recommended to use data from multiple years and incorporate more variables to enhance the comprehensiveness of the DEA analysis. Employing additional regression algorithms and optimization models from other heuristic algorithms is also suggested to further improve the objective function value.

Supplemental Information

Supplemental Information 1 Cluster and members.

Supplemental Information 2 Output-oriented DEA-CCR result members.

Supplemental Information 3 Minmax scaler result.

Supplemental Information 4 Cluster and centroid FCM.

Supplemental Information 5 Statistic descriptive.

Supplemental Information 6 Input-oriented DEA-CCR result members.

Supplemental Information 7 Log transformation scaler result.

Supplemental Information 8 Ideal dataset.

Supplemental Information 9 Historical dataset.

Supplemental Information 10 Columns list and description.

Supplemental Information 11 Statistical description.

Supplemental Information 12 Skewness.

Supplemental Information 13 Normalized Z-Score scaler result.

Supplemental Information 14 Minmax scaler result.

Supplemental Information 15 Log transformation scaler result.

Supplemental Information 16 Cluster and centroid from FCM.

Supplemental Information 17 Cluster and members.

Supplemental Information 18 Input-oriented DEA-BCC result members.

Supplemental Information 19 Ouput-oriented DEA-BCC result members.

Supplemental Information 20 Final Result of the DEA Process.

Supplemental Information 21 Columns list and description.

Supplemental Information 22 Historical data.

Supplemental Information 23 Normalized Z-Score scaler result.

Supplemental Information 24 Python code to develop a simulation app with comments to show the workflow.

Supplemental Information 25 Python code for developing app with comments to show the workflow.

Supplemental Information 26 Python code for developing app with comments to show the workflow.

Supplemental Information 27 Raw complete data.

Supplemental Information 28 Genetic Algorithm GBR code with comments to show the workflow.

Supplemental Information 29 Genetic Algorithm MLPR code with comments to show the workflow.

Supplemental Information 30 The result of dataset after its efficiency has been clustered.

Supplemental Information 31 Python code for processing DEA with comments to show the workflow.

Supplemental Information 32 Genetic Algorithm RFR code with comments to show the workflow.

Supplemental Information 33 Python code for developing an app in preprocessing the datasets.

Supplemental Information 34 Genetic Algorithm SVR code with comments to show the workflow.

Supplemental Information 35 List of all libraries versions required to run the system.

Supplemental Information 36 Python code for developing an app in clustering the data.

Supplemental Information 37 Python code for regression with comments to show the workflow.

Supplemental Information 38 DEA Dataset for benchmarking.

Supplemental Information 39 The first DEA dataset.

Supplemental Information 40 The second DEA dataset.

Supplemental Information 41 The third DEA datasets.

Supplemental Information 42 The mapping between the raw data and the masked datasets to be ready for analyzing.

Supplemental Information 43 Codebook.

Additional Information and Declarations

Competing Interests

The authors declare that they have no competing interests.

Author Contributions

Shofinurdin Soffan conceived and designed the experiments, performed the experiments, analyzed the data, performed the computation work, prepared figures and/or tables, authored or reviewed drafts of the article, and approved the final draft.

Arif Bramantoro conceived and designed the experiments, analyzed the data, prepared figures and/or tables, authored or reviewed drafts of the article, and approved the final draft.

Ahmad A. Alzahrani analyzed the data, authored or reviewed drafts of the article, and approved the final draft.

Data Availability

The following information was supplied regarding data availability:

The data and code are available in the Supplemental Files.

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
