# Peer review of "Combination of machine learning and data envelopment analysis to measure the efficiency of the Tax Service Office"

_PeerJ Computer Science, doi:10.7717/peerj-cs.2672_

## Round 0.1 · original submission · Major Revisions

Thank you for submitting your manuscript to PeerJ. After careful consideration of the reviewers' comments, we inform you that revisions are required before your paper can be considered further for publication.

The reviewers have raised several concerns. These issues need to be thoroughly addressed to ensure the robustness and validity of your findings.

In addition, we recommend a comprehensive rereading of your manuscript to correct any typographical, grammatical, or formatting errors that may have been overlooked.

Please revise your manuscript accordingly and provide a detailed response to each of the reviewers' comments. We look forward to receiving your revised submission and appreciate your understanding and cooperation in improving the quality of your work.

·

Basic reporting

In general, the authors demonstrated an adequate understanding of the researched matter and used appropriate literature to support their study. More importantly, they did a good job of presenting gaps in the literature and explaining how they plan to fill them.

Nevertheless, despite the obvious contribution of the paper, not enough is done to emphasize it. The only reference to the practical value of the developed methodology is made at the end of the Introduction. However, a single generic sentence is not enough to justify the publication of a paper in a renowned academic journal. More effort in this respect is hence required.

Although the quality of the language is generally satisfactory, proofreading is highly recommended. Besides a number of grammar mistakes, there are also many typos that could have been removed by careful re-reading of the text. One, but certainly not the only, example is the formula for MSE on page 6 (what is exactly y, and why does it not have a subscript?). In fact, most variables and indices in formulas throughout the manuscript remained explained. For instance, it is not clear what u’s and v’s on page 3 stand for.

Also, the paper is full of confusing phrases and terms. For instance, on page 10 the authors mentioned ‘ideal data’. It would be nice to know the meaning of this phrase, as well as why such data was needed in the first place.

On the other hand, there is no need to explain every single detail of the data-engineering phase. For instance, not only will everyone assume that the dataset was anonymized, but they will not even care about the masking process (Figure 4).

Experimental design

There are two salient problems with the experimental design, which need to be properly addressed. First of all, the dataset contains only 352 entities, which is by no means enough to train robust machine learning models. I know that not much can be done about the sample size in this particular case, but the authors should at least provide a discussion of how the limited sample size affected the final results. I also recommend that the authors provide references to other studies that relied on datasets of similar size.

In addition, it can be concluded (although it was not clearly stated) that the multilayer perceptron does not use any activation function. If this is true, what is the point of having five hidden layers? Since a linear combination of linear functions is again a linear function, hidden layers are fully redundant (i.e., the same result would have been achieved without any hidden layer). I recommend that the author familiarize themselves with the paramount role of activation functions in artificial neural networks and the repercussions of their absence.

Validity of the findings

I must admit I do not understand the purpose of the regression part of the analysis. From the section ‘Regression Process’, it can be concluded that all input and output variables are used as predictors in the machine learning models (actually, this is the most reasonable assumption, even though the authors most likely incorrectly labelled them as dependent variables). If this is indeed the case, would it not be easier and more accurate just to apply DEA to any new DMU? The regression models used in this study are typically applied to predict future outcomes and/or to address the incompleteness of data. If one has to have all the data in advance, what is the point of predicting efficiency scores when there is already a robust method for calculating these scores directly? I may be misunderstanding something here, but this definitely needs to be clarified.

In addition to this, the obtained values of MSE indicate the existence of severe bias in all models. I recommend that the author clarify what strategies they applied to prevent overfitting and to what extent these strategies were efficient. From the submitted .py files, it is obvious that this issue was taken into account, but for some reason, not a single word about this matter is said in the manuscript.

Finally, it is a good practice to include information about the version of Python and its modules used in the analyses (either in each .py file or in some supplementary document). It is hard to check and reproduce the results without this vital information.

Additional comments

This study has a lot to offer given the methodological novelties it brings. However, the positives are overshadowed by a number of methodological issues and the low efforts invested in ‘selling the final product’. While I cannot recommend this paper for publication in its present form, I am confident it has great potential. In line with that, I suggest 'Major Revisions'.

·

Basic reporting

The manuscript is written in a clear an unambiguous manner. Several typos were detected. For instance, the citation in line 33 lacks a word ‘Ministry’. The same stands for the list of references. The tense in line 215 should be changed to present continuous. Tables 2-4 should be positioned below the text explaining these tables for consistency of reading.
The literature used in this study is relevant and sufficient to explain the background of the analysis.
The article is well structured but lacks some important parts. First, it would be beneficial to have an explanation of the main contributions of the paper as early as in the introduction.

Experimental design

The analytical framework, and research questions are well defined, relevant and meaningful. I would advise the authors to provide further explanation on the following:
- Explicitly explain that the DMUs in the study were tax service offices. Also, is the total of 352 DMUs also a total population (the full list of tax service offices) or just a sample. This explanation is needed for the international readership of the journal. Do these offices have any autonomy/discretion in deciding on inputs used in the DEA analysis?
- Explain what was the main reason for the selection of this set of INPUT indicators? Some studies explicitly state that the ICT usage by tax administration, educational background of employees, experience of tax collectors might be meaningful predictors of tax administration performance [see Milosavljević, M., Radovanović, S., & Delibašić, B. (2023). What drives the performance of tax administrations? Evidence from selected European countries. Economic Modelling, 121, 106217. https://doi.org/10.1016/j.econmod.2023.106217]

- Explain what the main reason was for selecting this group of output indicators. For instance, indictors such as tax evasion crossed my mind. Additionally, why were the absolute measures used? Instead of using ‘Compliance rate of annual tax return submission’ the authors could have used ‘Compliance rate of annual tax return submission PER TAXPAYER’ to have comparable data for different.

As for the three comments listed above, I do not advise conducting additional analyses as this would require additional time and resources and certainly dilute the conclusion. Simple mentioning these limitations in the methodology would be sufficient.

Validity of the findings

The study is novel, and all the supplementary data provided alongside the manuscript.
Conclusions are well stated and linked to the original research questions. Nonetheless, I would advise authors to add few paragraphs in the conclusions related to the following:
- Comparison with what was assumed efficient prior to their study. Is there any analysis showing different or same findings as their study? Comparing such results would be highly beneficial for the fundamental understanding of the novelty of this paper.
- More important than previous, I would highly recommend to authors to add a paragraph on the practical implications of this study. Although it goes beyond the aims and scopes of the journal, policy recommendation with elements such as – what DMU can be considered as a role model, what input makes the most difference in the overall score – could profoundly change the practical contribution of this paper.

Additional comments

The paper provides a novel contribution both in methodological and practical areas. It was a great pleasure reviewing this paper.

---

## Round 0.2 · Minor Revisions

Thanks for your submission. After a careful review, a reviewer raised some other concerns about structure and methodology of the paper. Please revise the paper and submit an updated version with a point-by-point response letter. Although the re-review contains many comments, overall they should be minor to address.

·

Basic reporting

-

Experimental design

-

Validity of the findings

-

Additional comments

After addressing most of the comments, the final result is a better and more robust manuscript than was originally the case. However, in addition to a few neglected comments, there are also a few failed attempts to resolve the emphasised issues, which made me wonder whether you truly understand what you are doing and why you are doing this.

I do not mind giving you another try, but please be aware that I cannot let the paper through until everything is cleared out.

I will start with the most important issue, which is your justification of the regression part of the analysis. Specifically, in the section ‘Regression process’ (p. 12) you added the following sentence: “The DEA method can only produce efficiency values for existing tax service offices that have values for each variable, so the new tax service office cannot be included in the DEA model“.

However, in the very first sentence of the next paragraph, while describing the regression process, you said: “The independent variables (x) consist of the input and output variables, while the dependent variable (y) is the static efficiency value generated in the DEA process.”

So, which of these sentences is true? Both cannot be true since they are mutually contradicting.

Yes, the regression part is indeed essential because it addresses one important limitation of the DEA calculation procedure. However, this has nothing to do with the explanation you provided. You might want to read more about this matter, starting from the papers cited in the manuscript (e.g., those from Table 3).

This brings us to another important comment of mine, which you ignored for some reason. Once I started to examine your code line by line, I realised your analysis could not be replicated. Let me repeat once again that it is essential to include information about the versions of Python modules used in the analysis. Also, it is recommended to add comments in the code wherever possible and relevant so as to make it easier to follow the procedures. Most importantly, make sure that all key files are included! For instance, I cannot see the file named “hasil_efisiensi_klaster”, which is the starting point of the ML part. The dataset is anonymised, which means that nothing prevents you from sharing this and a few other missing files.

Turning to k-fold cross-validation, this should be specified and justified in the methodology (and additionally discussed together with other results), certainly not in the conclusion. Also, please include some other relevant statistics (e.g., standard deviation) that speak about the robustness of your model.

Moreover, be clear and transparent about your approach from the very beginning. For instance, in the ‘Methods’ section you said that the following four approaches will be used: Fuzzy C-Means; DEA; artificial neural networks; and Genetic Algorithm. However, it turned out that, besides ANN, you also used three additional ML techniques. These three suddenly came out of nowhere on page 12, without an explanation of why precisely these methods were used, as well as without any implementational details. On the other hand, the reader later finds out that the Genetic Algorithm was actually not used as a core technique but only as a hyperparameter tuning tool.

Speaking about methodology, make sure you fully understand the terms ‘multi-layer perceptron’ and ‘artificial neural network’. The same is true for the differences and similarities between ‘machine learning’ and ‘deep learning’. Once you master this, you will be able to avoid redundant terminology and remove incorrect statements. If nothing else, you will realise how absurd is the following sentence from page 6: “In this study, a ReLU activation function is used, although it is more commonly applied in deep learning, as shown by Bramantoro and Virdyna (2022).”

Finally, please address comments and recommendations from the previous round of the review that are still pending. While you do not have to answer positively to each and every of them, ignoring them is certainly not the solution.

·

Basic reporting

The authors have significantly improved the paper with regards to the language, structure and formal presentation of results.

Experimental design

The authors have significantly improved the methodology of the paper.

Validity of the findings

Findings are highly important with authentic novelty to the field.

Additional comments

The paper is acceptable for the publication in the current form.

---

## Round 0.3 · Minor Revisions

Thank you for submitting your manuscript to PeerJ Computer Science. After careful review, the reviewers have raised some concerns regarding the methodology and experimentation that need to be addressed before we can proceed with the publication.

We kindly request that you revise your manuscript in light of the reviewers' comments and make the necessary adjustments. Please also provide a detailed response letter addressing each of the reviewers' suggestions and observations.

We are confident that, with these revisions, your manuscript will be considered for publication.

Thank you again for your contribution, and we look forward to receiving your revised submission.

·

Basic reporting

-

Experimental design

-

Validity of the findings

-

Additional comments

The positive thing is that most of my comments from the previous two rounds have been addressed. However, a few essential ones are still unresolved. More importantly, your attempts to clarify some of the emphasised issues further exposed an insufficient level of knowledge about some aspects of the explored matter.

While most reviewers would easily reject the paper based solely on these evident knowledge gaps, I believe it is worth trying one more time. However, please note there will be no fifth round of the review.

To simplify things, this time I left comments directly in the manuscript. The only exemption concerns the justification of the regression stage of the study, which is elaborated here. To avoid any confusion (which, apparently, was present previously), I tried to be as comprehensive as possible in my reasoning.

So, let me now focus on the justification of the regression. In the previous two waves, I communicated clearly that your arguments based on the establishment of new units do not hold. Since you decided to stick to your original story, I will now do my best to demonstrate its fallacy.

Specifically, your estimation procedure is divided into three stages:

1) clustering of DMUs based on 7 input variables
2) DAE based on all 13 input and output variables
3) regression based on all 13 input and output variables

So, to predict efficiency values for any DMU in the future (including the new ones), you need to have data for all 13 variables. I am not going into the discussion about the rationality and economic justification of guestimating some of those 13 values for the tax units that have not yet been established. Instead, let us just assume that we can come up with some credible numbers. However, once you have all 13 inputs, you have all the ingredients needed for stages 1) and 2) as well. So, the question I posed in the previous two rounds, which you failed to answer, is the following:

Why would someone rely on regression in a situation where it is methodologically more accurate to repeat clustering and DEA with the augmented dataset (i.e. existing DMUs + one or more units that are yet to be established)?

I was hoping that, while thinking about this, you will also figure out that regression can be quite useful even in the case of the existing DMUs. I did give you a hint by stating that some of the papers cited in your manuscript actually provide a clear answer to this question. Since you ignored this suggestion, I decided to copy what Zhu et al (2021) said on this. Specifically, on p. 436 there is the following paragraph:

”During evaluating organizational performance, however, if a new DMU needs to be known its efficiency score, the DEA analysis would have to be re-conducted. Especially, nowadays DMU datasets are growing quickly in the real world with the rapid development of big data. For example, the number of small and micro-sized companies in mainland China has exceeded 73 million, and the number is still increasing (Zhang, 2017). Hence, when we have already calculated DEA efficiency for a large number of DMUs, and if a new DMU needs to be known its efficiency score, the DEA model would have to be re-run, which would need huge computer resources in terms of memory and CPU time. Therefore, we attempt to propose a way to predict the efficiency score using machine learning (ML) algorithms“.

It could be that their focus on new DMUs confused you. However, what Zhu et al (2021) failed to emphasise, and the same applies to you, is that regression can be applied to the existing DMUs as well, once new data for them becomes available (i.e., in the subsequent years). This would say that the valid justification for the reliance on regression has nothing to do with whether a DMU already exists or not, but simply with time and resource management. This is not overly relevant in your case since you have only 352 DMUs, but it does become extremely important when dealing with millions of units.

Another valid argument, which, however, not only requires additional justification but could raise some questions about the real value of DEA, resides in the fact that steps 1) and 2) repeated on an augmented dataset would yield different numbers for the existing DMUs. Given this, I suggest that you stick to the story by Zhu et al (2021).

Also, I repeat that I left a number of other comments in the manuscript.

---

## Round 0.4 · accepted · Accept

I hope this message finds you well. After carefully reviewing the revisions you have made in response to the reviewers' comments, I am pleased to inform you that your manuscript has been accepted for publication in PeerJ Computer Science.

Your efforts to address the reviewers’ suggestions have significantly improved the quality and clarity of the manuscript. The changes you implemented have successfully resolved the concerns raised, and the content now meets the high standards of the journal.

Thank you for your commitment to enhancing the paper. I look forward to seeing the final published version.

[# ·

Basic reporting

-

Experimental design

-

Validity of the findings

-

Additional comments

I do not have any further comments. The paper can be published as it is.